37

REVIEW-SYMPOSIUM

# The ventromedial prefrontal cortex and emotion regulation: lost in translation?

Laith Alexander[1,2] (iD), Christian M. Wood[3,4] (iD) and Angela C. Roberts[3,4] (iD)

[1]*St Thomas' Hospital, London, UK*
[2]*Department of Psychological Medicine, School of Academic Psychiatry, Institute of Psychiatry, Psychology and Neuroscience, King's College London, London, UK*
[3]*Department of Physiology, Development and Neuroscience, University of Cambridge, Cambridge, UK*
[4]*Behavioural and Clinical Neuroscience Institute, University of Cambridge, Cambridge, UK*

Edited by: Ian Forsythe & Michael Okun

The peer review history is available in the Supporting information section of this article (https://doi.org/10.1113/JP282627#support-information-section).

**Abstract**   Neuroimaging studies implicate the ventromedial prefrontal cortex (vmPFC) in a wide range of emotional and cognitive functions, and changes in activity within vmPFC have been

---

This symposium review forms part of the 'Decoding Prefrontal Cortical Physiology: Circuits of Cognition' symposium held at Physiology 2021 in July 2021, and organised by Professor Matt Jones.

---

*The Journal of Physiology*

linked to the aetiology and successful treatment of depression. However, this is a large, structurally heterogeneous region and the extent to which this structural heterogeneity reflects functional heterogeneity remains unclear. Causal studies in animals should help address this question but attempts to map findings from vmPFC studies in rodents onto human imaging studies highlight cross-species discrepancies between structural homology and functional analogy. Bridging this gap, recent studies in marmosets – a species of new world monkey in which the overall organization of vmPFC is more like humans than that of rodents – have revealed that over-activation of the caudal subcallosal region of vmPFC, area 25, but not neighbouring area 32, heightens reactivity to negatively valenced stimuli whilst blunting responsivity to positively valenced stimuli. These co-occurring states resemble those seen in depressed patients, which are associated with increased activity in caudal subcallosal regions. In contrast, only reward blunting but not heightening of threat reactivity is seen following over-activation of the structurally homologous region in rodents. To further advance understanding of the role of vmPFC in the aetiology and treatment of depression, future work should focus on the behaviourally specific networks by which vmPFC regions have their effects, together with characterization of cross-species similarities and differences in function.

(Received 30 November 2021; accepted after revision 13 May 2022; first published online 30 May 2022)

**Corresponding author** A. C. Roberts: Department of Physiology, Development and Neuroscience, Downing Street, Cambridge CB2 3DY, UK. Email: acr4@cam.ac.uk

**Abstract figure legend** The value of non-human primates in bridging the translational divide between correlative functional neuroimaging studies of the ventromedial prefrontal cortex (vmPFC) in humans and intervention studies of prelimbic (PL) and infralimbic (IL) cortex in rodents with a view to understanding the aetiology and treatment of mood and anxiety disorders.

## Heterogeneity within human ventromedial prefrontal cortex

Neuroimaging studies have implicated the ventromedial prefrontal cortex (vmPFC) in a range of functions that include not only emotion regulation and generation but also action control, memory and economic decision making (reviewed in Hiser & Koenigs, 2018). Recognition of its importance in mood and anxiety disorders comes from reported reductions or increases in activity within vmPFC associated with their symptomatology (Drevets et al., 1997; Hiser & Koenigs, 2018; Morey et al., 2015). In relation to major depressive disorder (MDD) such

changes have been linked not only to the aetiology of MDD but also to successful treatment response (Mayberg et al., 2000). Indeed, deep brain stimulation of this region itself has become a potentially successful treatment of refractory depression (Mayberg et al., 2005), and activity change within it (Downar et al., 2014; Fox et al., 2012; Hiser & Koenigs, 2018; Salomons et al., 2014) as well as its inter-regional functional connectivity (Dunlop et al., 2017; Weigand et al., 2018) has been identified as a good predictor of response to a variety of treatments.

However, the vmPFC is a large, structurally heterogeneous region composed of several sub-regions. As identified in architectural maps of Petrides et al.

**Laith Alexander, MBBChir PhD**, is an Academic Foundation Doctor in Psychiatry at King's College London, having completed his PhD at the University of Cambridge exploring the role of the ventromedial prefrontal cortex in aversive learning, appetitive learning and cardiovascular regulation. **Christian M. Wood, PhD**, is a Postdoctoral Research Associate working with Professor Roberts at the University of Cambridge, and the focus of his current work is understanding the ventromedial prefrontal cortex pathways that underlie symptoms and successful treatment in mood and anxiety disorders. **Angela C. Roberts, PhD, FMedSci**, is a Professor of Behavioural Neuroscience at the University of Cambridge. Her research focuses on the executive control functions of the prefrontal cortex, their contribution to emotion regulation, how they are instantiated in cortico-subcortical circuits and their hierarchical integration.

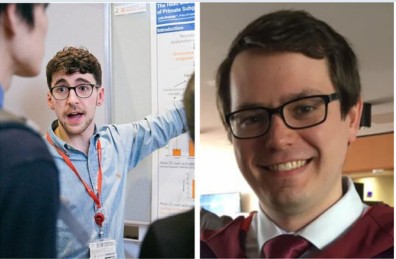
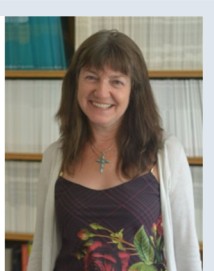

(2012), Ongür et al. (2003) and Palomero-Gallagher et al. (2015), this region variably includes area 25; the subgenual portions of areas 32 and 24; and areas 14 and 10. Of note are the regional differences in the maps of Ongür et al. and Petrides et al., particularly with regards the extent of areas 14 and 10 on the caudal aspects of the medial surface, and the extension (or not) of area 25 onto the orbital surface. This further complicates the determination of homologies between species. Anterograde and retrograde tracing studies in macaque monkeys (An et al., 1998; Carmichael & Price, 1995; Joyce & Barbas, 2018; Kondo et al., 2003, 2005; Ongür et al., 1998; Saleem et al., 2008) and more recently the marmoset (Majka et al., 2020; Ríos-Flórez et al., 2021) reveal marked differences in the cortical and subcortical connectivity patterns of these regions which likely underpin their functional heterogeneity.

Synthesizing results across several imaging studies, recent meta-analytical data reveal at least two, if not three, functional subdivisions within vmPFC: two anterior pregenual regions related to value/decision making and social cognition, respectively, and a posterior post-(sub)genual (or subcallosal) region related to emotion regulation and generation (Hiser & Koenigs, 2018). A database-driven analysis of the subcallosal region by Palomero-Gallagher and colleagues including areas 25, subgenual area 24 and subgenual area 32 highlighted differences between these regions both in functional connectivity and behavioural domains (Palomero-Gallagher et al., 2015). In this analysis, both subgenual 24 and subgenual area 32 were associated with cognitive and emotional domains – however, whilst subgenual area 24 was associated with sadness (and taste), subgenual area 32 was associated with fear. Only area 25 did not appear to be associated with a behavioural domain although its functional connectivity co-activated selectively with an autonomic afferent processing network. Consideration should be given, though, to the positioning of this entire region above the orbits and the sinuses, making it prone to imaging artifacts and making localization of function within this region using neuroimaging particularly challenging (Du et al., 2007; Volz et al., 2019).

Causal studies in animals, in which selective interventions of specific regions within vmPFC are investigated, should provide insight into these proposed functional subdivisions and their relationship to symptoms of MDD. But so far, progress has been hampered, in part by difficulties in determining cross-species anatomical and functional equivalence. This review sets out to consider some of the issues that are currently confounding the effectiveness of translational studies in the vmPFC. It first considers cross-species anatomical similarities and discrepancies within this region. It then focuses on one aspect of emotion regulation – the extinction of conditioned threat

responses – that has been directly compared across studies of the vmPFC in humans, the common marmoset and rodents. Discrepancies arising from these comparisons are highlighted and discussed. A series of studies in the marmoset monkey are then summarized that further our understanding of this important region in relation to anhedonic and comorbid anxiety symptoms of MDD. We complete the review by highlighting lessons learnt that will facilitate the future of translational studies within vmPFC.

## Lost in translation: anatomical similarities and discrepancies

It has been hypothesized that anatomically homologous regions within vmPFC are present across rodents, monkeys and humans which should afford the opportunity to establish causal evidence for functional heterogeneity suggested by the neuroimaging literature in humans. It is important to recognize, though, that whilst the term vmPFC is commonly used to refer to the entire ventromedial frontal lobes, only more rostral regions are specifically prefrontal. More caudal aspects are part of the cingulate cortex, and the proposed anatomical cross-species homology relates primarily to these cingulate regions of 24, 25 and 32.

Areas 24, 25 and 32 have been identified in monkeys and rodents (Vogt, 2016; Vogt & Paxinos, 2014) although both areas 24 and 32 have more subdivisions in humans and monkeys than in rodents. In rodents, these regions map onto the more commonly described areas of anterior cingulate (AC1/Cg1 and AC2/Cg2), prelimbic (PL, Cg3) and infralimbic (IL) cortices, respectively (Myers-Schulz & Koenigs, 2012; Vogt & Paxinos, 2014). Ventral PL and IL are often referred to as vmPFC and compared with vmPFC in humans and primates. Based on cytoarchitectonics, IL is proposed to be homologous to primate area 25, whereas PL is proposed to be homologous to primate area 32 (Fig. 1).

Consistent with their respective homologies, there are marked similarities in the connectivity patterns of rodent IL and non-human primate area 25, and rodent PL and non-human primate area 32. In the case of IL and area 25, connectivity with regions linked to emotion, visceromotor control and memory is broadly similar. Subcortically, this includes the hypothalamus, amygdala, bed nucleus of the stria terminalis (BNST), lateral septum, nucleus accumbens, midline thalamic nuclei, periaqueductal grey (PAG) and parabrachial nucleus; and cortically, this includes the medial orbitofrontal cortex (OFC), insula, anterior cingulate, perirhinal cortex, entorhinal cortex, hippocampal formation and PL/area 32 (Joyce & Barbas, 2018; Vertes, 2004). Regarding PL and area 32, these regions project to the amygdala, hypothalamus, medial

caudate and nucleus accumbens subcortically; and the agranular insular, other cingulate subregions and OFC cortically (Chiba et al., 2001; Yeterian et al., 2012).

However, cross-species differences in connectivity do exist, especially when considering cortico-cortical connectivity. Absent in rodent IL, but present in macaque area 25, are connections with auditory association and polymodal sensory cortex within the superior temporal gyrus (Hoover & Vertes, 2007; Vertes, 2004), In addition, rodent IL lacks the prominent connections that primate area 25 has with higher-order associations areas within PFC, including area 9, 10 and 46 and posterior cingulate area 23, regions with no known anatomical homology in rodents (compare Yeterian et al. (2012) and Joyce & Barbas (2018) with Vertes (2004) and Hoover & Vertes (2007)).

In addition, even subcortically where there is considerable comparability, there are notable differences in the extent and the specificity of connectivity within regions. First, macaque area 25 neurons project primarily to central and basal amygdala nuclei. In contrast, a major

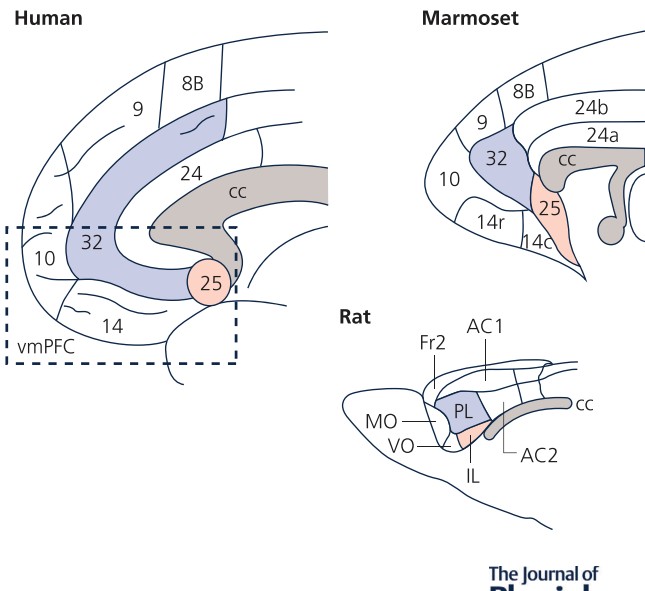

**Figure 1. Anatomical homology across human, marmoset and rodent ventromedial prefrontal cortex (vmPFC) and cingulate cortex**
The term vmPFC typically refers to the entire ventromedial frontal lobe in humans. However, whilst more rostral regions are neocortical, more caudal regions such as areas 24, 25 and 32 consist of allocortex and form part of the cingulate gyrus. It is to these areas that the proposed homology typically relates. Human area 25 and non-human primate (marmoset, macaque) area 25 is proposed to be anatomically homologous with rodent infralimbic cortex (IL). Human area 32 and non-human primate area 32 is thought to be anatomically homologous with rodent prelimbic cortex (PL). Cytoarchitectonic divisions based on Petrides et al. (2012) (human), Paxinos et al. (2011) (marmoset) and Palomero-Gallagher & Zilles (2015) (rat).

projection of IL is onto the GABAergic inhibitory inter-calated cell masses (proposed to mediate the inhibitory effects of IL stimulation on conditioned threat responses – see below), which is more comparable to connections of posterior OFC in macaques (Joyce & Barbas, 2018). Second, connectivity of primate area 25 is more extensive and widespread across the hypothalamus compared to that of rodent IL (Ongür et al., 1998; Rempel-Clower & Barbas, 1998; Vertes, 2004), with many more hypothalamic neurons sending reciprocal projections back to macaque area 25. Third, in the brainstem, primate area 25 projects predominantly to dlPAG whilst rodent IL projects more evenly to both vlPAG and dlPAG; vlPAG is associated with passive coping responses, dlPAG with active coping responses (An et al., 1998; Floyd et al., 2000). Moreover, absent in macaque area 25, but prominent in rodent IL, are direct projections to autonomic effectors including the nucleus of the solitary tract, ventrolateral medulla and intermediolateral cell column (Joyce & Barbas, 2018). Area 25 is likely to have effects instead indirectly via the parabrachial nucleus and forebrain sites including the hypothalamus and BNST.

Like IL and area 25, differences between PL and area 32 are most striking when considering their cortical connectivity patterns. First, pgACC-32 has far greater connectivity with lateral prefrontal association areas (8, 9, 46 and 47), premotor area 6 as well as posterior cingulate regions, 23 and 31, and retrosplenial cortices, 29 and 30, than neighbouring sgACC-25, but aside from premotor and retrosplenial cortex, these regions have no known structural homologues in rodents (again comparing Chiba et al. (2001) and Yeterian et al. (2012) with Hoover & Vertes (2007) and Vertes (2004)). Like sgACC-25, pgACC-32 also has extensive connections with the auditory association and multimodal sensory regions of the STG, projections that are absent in rodent PL; these temporal regions projecting instead to more dorsal cingulate regions in rodents (Hoover & Vertes, 2007).

Thus, whilst there is considerable overlap in connectivity between rat IL and monkey area 25, and rat PL and monkey area 32, the cortical connectivity patterns of these two brain regions in macaque (which have been largely corroborated in marmosets; Majka et al., 2020; Ríos-Flórez et al., 2021) set them apart from their proposed rodent counterparts. Consequently, whilst they may be involved in similar subcortical functional networks across species, the precise role they play may well have altered.

## Lost in translation: functional similarities and discrepancies

A series of seminal studies in rats revealed the opposing roles of PL and IL in the regulation of conditioned

threat responses. Inactivation of IL prior to extinction of a Pavlovian conditioned threat response disrupted the extinction of this response as well as extinction recall the next day (Sierra-Mercado et al., 2011). In contrast, inactivation of PL reduced the initial recall of the conditioned threat response at the start of the extinction procedure. Electrophysiological, pharmacological and stimulation studies have since all supported the hypothesis that IL facilitates context-specific suppression of conditioned threat responses whilst PL maintains such threat responses (Milad & Quirk, 2012).

Subsequent studies in humans, in some cases using an almost identical paradigm, have identified changes in activity in human vmPFC related to extinction recall, consistent with findings following IL manipulations (Dunsmoor et al., 2019; Milad, Wright et al., 2007; Phelps et al., 2004) and alterations in activity in dorsal anterior cingulate cortex more consistent with findings following PL manipulations (Milad, Quirk et al., 2007). However, closer inspection of these parallel sets of findings reveals inconsistencies. Whilst IL is purported to be equivalent to area 25, the extinction recall-related activation in humans is far more rostral to human area 25, being at the level of and in front of the genu, in what is identified as area 14 based on the atlas of Petrides et al. (2012) and area 10 based on the atlas of Ongür et al. (2003). Moreover, although PL is purported to be equivalent to area 32, it was a region in human dorsal anterior cingulate cortex (area 24; mid cingulate cortex, as defined by Vogt (2016)) that appeared consistent with the PL involvement in the facilitation of conditioned threat expression. These disconnects between anatomical homology and function when translating studies between animals and humans require attention if we are to understand the complex neural processes involved in emotion regulation and expression. Despite the incredible advance in technologies to study cortical functioning over the past 15 years, there still exists this fundamental knowledge gap of the causal role of individual vmPFC regions in these emotional processes.

One approach to addressing these discrepancies and to bridge this apparent gap between rodent and human studies is to determine the contribution of these regions of vmPFC in a non-human primate in which the overall structural organization of vmPFC is far similar to that of humans than that of rodents (Ongür & Price, 2000; Palomero-Gallagher & Zilles, 2015; Paxinos et al., 2011). Already the use of anatomical tracing (Chiba et al., 2001; Freedman et al., 2000; Joyce & Barbas, 2018; Ongür & Price, 2000) and electrophysiological recordings (Amemori & Graybiel, 2012; Monosov & Hikosaka, 2012) in macaques has begun to highlight the heterogeneity in connections and functional representations within the vmPFC. In the next section, we describe a series of intervention studies determining the contribution of regions of vmPFC in a new world monkey, the common marmoset, in the regulation of both negative and positive emotion.

## Can primates bridge the human–rodent gap?

By combining targeted intracerebral infusions with remote cardiovascular and behavioural monitoring in a new world monkey, the common marmoset, it has been determined how subregions of non-human primate vmPFC differentially regulate physiology and behaviour. The combination of both behavioural and physiological measurements maximized forward translation to humans – whose measures of emotion are predominantly physiological or subjective – together with back translation to rodent studies – whose measures are primarily behavioural. Owing to the apparently discrepant findings between rodent and human studies, the importance of non-human primate studies as a translational step towards understanding the function of human vmPFC is nowhere more evident than when addressing its role in emotion regulation.

In the first such study, issues of anatomical homology and functional analogy were addressed directly by infusing the $GABA_A/GABA_B$ receptor agonists muscimol/baclofen to temporarily inactivate marmoset caudal subgenual ACC (sgACC, corresponding to area 25; sgACC-25) and perigenual ACC (pgACC, corresponding to area 32; pgACC-32), and assessing the effects on autonomic regulation and threat-related behaviour (Wallis et al., 2017). In an emotionally neutral context (a testing chamber in which marmosets had been fully habituated), inactivation of sgACC-25 had profound effects on basal cardiovascular activity, increasing cardiac vagal tone and heart rate variability (reflecting dynamic adjustments in heart rate mediated by the parasympathetic and sympathetic nervous systems) and reducing heart rate. Inactivation of pgACC-32 had comparatively modest effects, leading to a small increase in blood pressure.

The profound consequences of sgACC-25 inactivation on resting autonomic tone have apparent, albeit tenuous, similarity to manipulations of its rodent counterpart. Rodent IL (and ventral PL) has been termed the 'visceral motor cortex' with projections well placed to alter autonomic tone, including to the hypothalamus, amygdala, insula and periaqueductal grey (Loewy & Spyer, 1990; Vertes, 2004). Both inactivation and activation of IL have cardiovascular effects, although the directionality of these effects is variable (Hassan et al., 2013; Loewy & Spyer, 1990; Müller-Ribeiro et al., 2012; Tavares et al., 2009; Terreberry & Neafsey, 1983). Macaque vmPFC projects extensively to regions involved in the central regulation of autonomic function (Barbas et al., 2003; Ongür & Price 2000) and work by Kaada and colleagues showed that direct electrical stimulation of macaque pgACC and

sgACC induced cardiovascular and respiratory changes (Kaada et al., 1949). These were most pronounced following stimulation of caudal regions corresponding to sgACC-25. However, electrical stimulation can impact on axons passing through the targeted region and so the localizability of these findings is unclear. Selective stimulation and inactivation of cell bodies in marmosets provides definitive evidence for the specific involvement of sgACC-25 in autonomic regulation and maps closely onto results from human neuroimaging studies exploring the neural correlates of heart rate variability. These studies report inverse correlations between human sgACC-25 BOLD activity with changes in parasympathetic tone during affective state switching (Lane et al., 2013), supporting the profound effects of marmoset sgACC-25 inactivation to increase cardiac vagal tone.

Any apparent functional analogy between marmoset and rodent sgACC-IL and pgACC-PL was not evident, however, when comparing their contribution to responsivity in negatively valenced contexts, specifically during Pavlovian conditioned threat discrimination and conditioned threat extinction (Wallis et al., 2017). If marmoset sgACC-25 were to be functionally analogous to rodent IL, inactivations of the former would be expected to impair extinction recall and potentially the extinction of conditioned threat responses too (Sierra-Mercado et al., 2011). In contrast, if pgACC-32 were to be functionally analogous to rodent PL, its inactivation would be expected to reduce the expression of conditioned threat and accelerate extinction of the threat response.

The precise opposite was found. Inactivation of sgACC-25 reduced both the behavioural ('vigilant scanning') and cardiovascular correlates of conditioned threat during Pavlovian conditioned threat discrimination and accelerated the extinction of conditioned behavioural and cardiovascular responses during threat extinction. This would suggest that an 'active' marmoset sgACC-25 promotes threat-congruent behavioural and cardiovascular responses, contrasting to an 'active' rodent IL which acts to diminish threat-congruent responses. Inactivation of pgACC-32 also had opposite effects to those that would be expected from inactivation of its purported rodent analogue, PL, causing threat generalization to the safety cue and to the overall context during aversive Pavlovian discriminative conditioning, together with impaired extinction of the threat response. It is worth noting, though, that in a recent study in mice, selective inactivation of projections from IL to the lateral septum did promote threat-congruent responses, similar to that seen following global inactivation of area 25 in marmosets (Chen et al., 2021). If replicated, this would suggest that IL in rodents can promote conditioned threat reactivity in specific contexts.

To further assess primate sgACC-25 in regulating responsivity to threat, outside of the Pavlovian domain, an approach-avoidance decision-making task was employed, in which marmosets performed operant responses for rewards on a touchscreen with the potential for unpredictable punishment (Wallis et al., 2019). Inactivation of sgACC-25 reduced punishment avoidance, concordant with evidence in the Pavlovian domain that this manipulation reduces the impact of threat (summarized in the lower panel of Fig. 2) and inconsistent with the consensus that rodent IL is involved with the suppression of threat responses.

In summary, these studies in primates do not support the proposed functional analogy between rodent IL and PL and primate areas 25 and 32, respectively – at least not in the domain of the regulation of conditioned threat responses.

## How strong are the monkey links to human psychiatric disorders?

So far, we have highlighted discrepancies between rodents and humans and between rodents and marmosets, but is there consistency between marmosets and humans? To maximize forward translation and construct validity to individuals with depression, we over-activated sgACC-25 to parallel the over-activity of the extended caudal subcallosal region reported in correlative human neuroimaging studies (Keedwell et al., 2010; Mayberg et al., 2005). We used two pharmacological methods: a combination of $mGlu_{2/3}$ and $GABA_B$ receptor antagonists (CGP-52432 and LY-341495) to enhance presynaptic glutamate release, and an inhibitor of the excitatory amino acid transporter-2, dihydrokainic acid (DHK), to reduce glutamate reuptake in astrocytes (thereby prolonging and potentiating glutamate effects at the synapse). Both effects are temporary, lasting approximately 20−30 min.

In an emotionally neutral context, over-activation of sgACC-25 enhanced cardiac sympathetic tone, reduced cardiac vagal tone, reduced heart rate variability and increased heart rate at rest (Alexander et al., 2020) – effects that were opposite to those of inactivation. In contrast, and highlighting the apparent selective role of area 25 within vmPFC in regulating basal cardiovascular activity, over-activation of neighbouring pgACC-32 and the more rostral area 14 (Stawicka et al., 2020) had minimal effect on basal cardiovascular function. These effects of sgACC-25 over-activation mirror the reduced parasympathetic tone often reported in mood disorders (Hartmann et al., 2019) and are relevant to the known association between mood disorders and cardiovascular disease (Davidson, 2012).

During aversive Pavlovian discriminative conditioning, over-activation of sgACC-25 induced generalization of cardiovascular and behavioural conditioned arousal to the context, and delayed extinction of conditioned threat responses (Alexander et al., 2020). Consistent

with heightened threat reactivity, over-activation of sgACC-25 led to increased punishment avoidance on an approach-avoidance decision-making task (Wallis et al., 2019), which mirrors the increased sensitivity to negative feedback observed in depressed patients (Elliott et al., 1996; Murphy et al., 2003; Roiser & Sahakian, 2013).

To determine whether over-activity within sgACC-25 was primarily involved in heightening responsivity to immediate threat or was also engaged in more uncertain, anxiety-provoking contexts, responsivity on the human intruder (HI) paradigm was investigated. Here, marmosets are confronted with an uncertain threat in the form of an unfamiliar human whilst in their home cage. Uncertain threat is proposed to lie more distally on the threat imminence continuum, and because the threat is further away (in time, space, or probability) it is hypothesized that there is more time to engage higher-order cognitive mechanisms to assess the situation (Fanselow & Lester, 1988; Mobbs et al., 2009, 2015, 2020; Perusini & Fanselow, 2015). sgACC-25 over-activation resulted in greater intolerance of this uncertain threat (Alexander et al., 2020) as measured by an increased anxiety score derived from an exploratory factor analysis of marmosets'

individual behaviours to this threat (described in Quah et al., 2020). If, as correlative human neuroimaging studies would suggest, sgACC over-activity is associated with anxiety and low mood (Drevets et al., 2008), these results demonstrate that this region may be causally involved, and that over-activity within sgACC-25 of the vmPFC results in an apparently overall negative emotional state (summarized in the upper panel of Fig. 2). In mice and rats the effects of manipulations of IL on tests of uncertain threat are mixed, IL being shown to promote (Bi et al., 2013; elevated plus maze, open field and novelty suppressed feeding), have no effect (Suzuki et al., 2016; open field) or diminish (Gasull-Camós et al., 2017; novelty suppressed feeding) reactivity.

A prominent symptom of depression is anhedonia, defined as a loss of the ability to experience pleasure. Anhedonia is a symptom construct which includes anticipatory (Pavlovian), motivational (operant) and consummatory (hedonic) components together with impairments in reward learning and decision making (Der-Avakian & Markou, 2012; Halahakoon et al., 2020; Treadway & Zald, 2011). We determined the possible roles of sgACC-25 and pgACC-32 in behavioural and

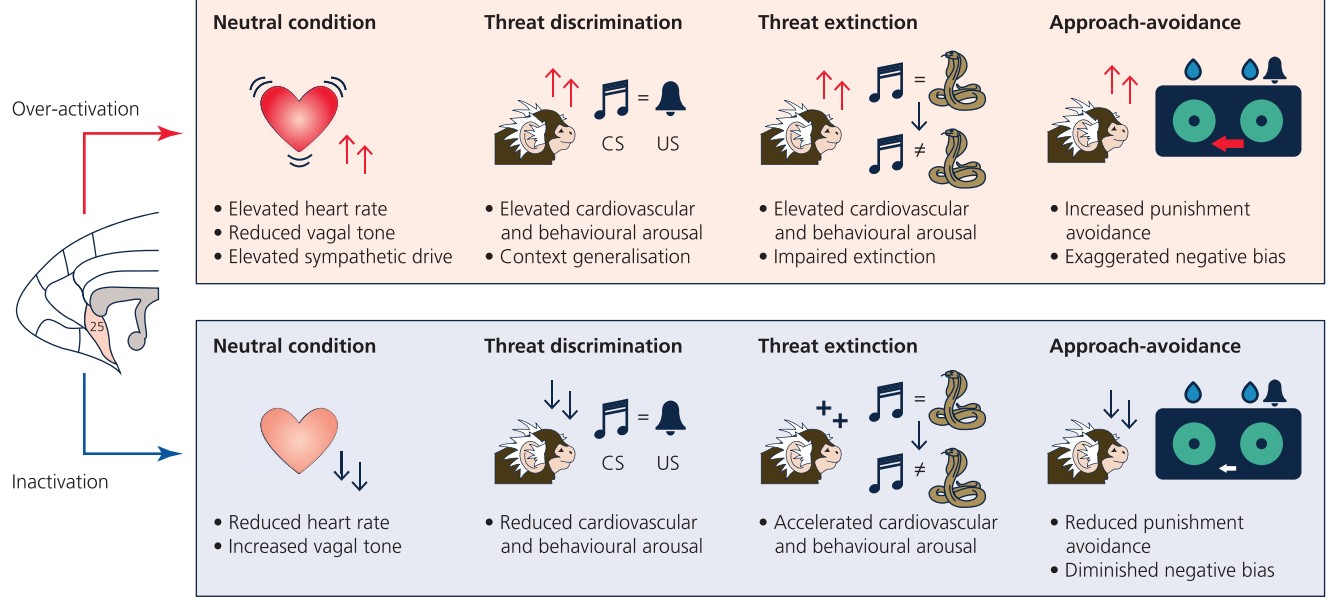

**Figure 2. Over-activation and inactivation of marmoset sgACC-25 have broadly opposite effects on cardiovascular function and threat responsivity**
Over-activation of sgACC-25 (top panel) results in a state of sympathetic over-activity, with elevations in heart rate, together with elevated cardiovascular and behavioural arousal in threatening contexts such as those in Pavlovian threat discrimination and threat extinction paradigms. In an operant approach-avoidance task, sgACC-25 over-activation increases punishment avoidance manifesting as an increased bias away from the punished side. Inactivation of sgACC-25 (bottom panel) induces a contrasting state, with increased resting vagal tone and reduced heart rate; reduced arousal during Pavlovian threat discrimination; enhanced rates of threat extinction; and reduced punishment avoidance on the approach-avoidance task.

cardiovascular impairments relevant to the anticipatory, motivational and consummatory forms of anhedonia. In an appetitive Pavlovian task, over-activation of marmoset sgACC-25 (but not pgACC-32) blunted correlates of reward anticipation but did not alter the amount of reward consumed or consummatory autonomic arousal (Alexander et al., 2019). To confirm that there were no effects of sgACC-25 over-activation on reward consumption, marmosets were also tested on the sucrose preference test – a classic rodent paradigm to assess anhedonia-like behaviour – where consumption remained intact. In the motivational domain, sgACC-25 over-activation reduced marmosets' willingness to work for reward, as reflected by the reduction in the breakpoint (maximum number of responses made to acquire reward) on an operant progressive ratio schedule of reinforcement, where animals must make more and more responses to receive reward (summarized in Fig. 3).

This dissociation between blunted reward anticipation and motivation, whilst reward consumption remains unaffected, suggests that sgACC-25 over-activation provides face validity to the blunted reward processing observed in the clinical state: depressed patients experiencing anhedonia exhibit similar blunting in the former two constructs but less so in the latter (Amsterdam et al., 1987; Arrondo et al., 2015; Berlin et al., 1998; Dichter et al., 2010; Klein, 1987; McFarland & Klein, 2009; Treadway & Zald, 2011; although see McCabe et al. (2012) for neural changes during reward consumption in an at-risk group). This over-activation induced blunting of reward-related responses, alongside enhanced threat-related responses, reveals the domain specificity of the negative affective state, as distinct from an overall state of hyperarousal.

At this point, it should be noted that unlike the contrasting effects between sgACC-25 and rodent IL manipulations on the regulation of Pavlovian conditioned threat responses, there is some correspondence on the regulation of appetitive responses. DHK-induced over-activation of IL in rats blunts appetitive responses as measured by intracranial self-stimulation (John et al., 2012), broadly consistent with the effects of sgACC-25 over-activation in marmosets. Infusions of *N*-methyl-D-aspartate into IL reduce ventral dopamine neuronal firing (Moreines et al., 2017) and optogenetically induced increases in IL excitability block dopamine-induced place preference (Ferenczi et al., 2016; although see Fuchikami et al. (2015) for opposing effects). Potential differences in effects (Ferenczi et al., 2016; Fuchikami et al., 2015) may depend upon the time frame of the physiological manipulation (minutes, hours, etc.) and whether the effects are measured at the time of the

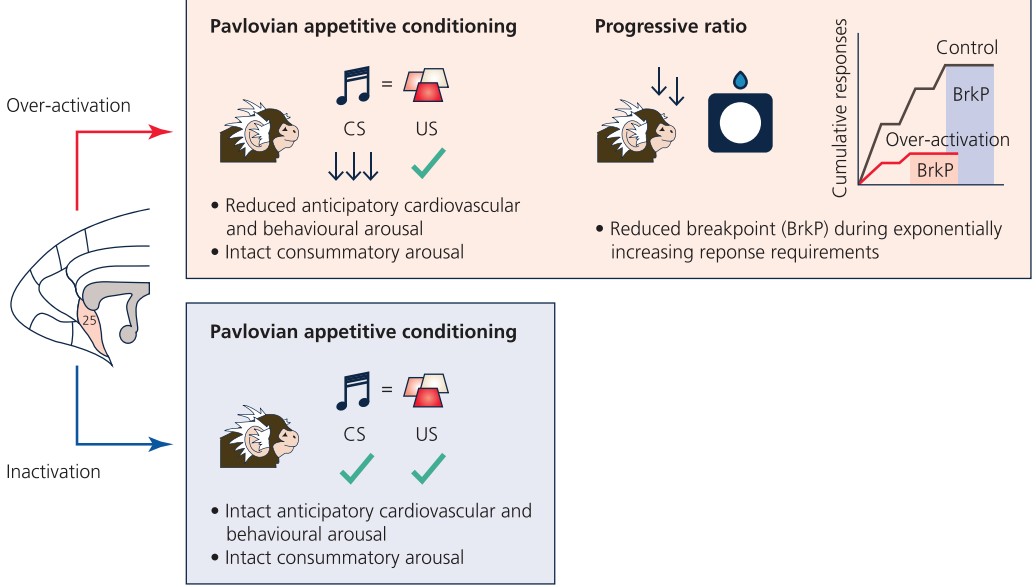

**Figure 3. Over-activation of marmoset sgACC-25 blunts reward anticipation during Pavlovian appetitive discrimination and reduces the willingness to work for reward**
Over-activation of sgACC-25 reduces the anticipatory (CS-induced) cardiovascular and behavioural arousal but has no effect on the consummatory (US-induced) cardiovascular or behavioural response in a Pavlovian appetitive discrimination task. It also reduces the extent a marmoset is willing to display increasing levels of responding for reward on a progressive ratio task. Inactivation of sgACC-25, by contrast, had no effect on either anticipatory or consummatory arousal.

manipulation or 24 h later. This needs to be borne in mind for future comparisons within and across species.

To explore the predictive validity of sgACC-25 manipulation in marmosets, we investigated whether the rapidly acting antidepressant ketamine could reverse any of the over-activation induced impairments in a time-dependent manner (Alexander et al., 2019, 2020). We chose ketamine owing to its rapid effects after a single dose, with particular efficacy for reward-related impairments (Lally et al., 2014, 2015).

A single intramuscular injection of ketamine (at a clinically relevant dose of 0.5 mg kg$^{-1}$) successfully reversed the blunted reward anticipation when sgACC-25 was over-activated at 1 day and 7 days following ketamine administration; the ameliorative effect subsided at 21 days post-administration. This time course is a striking parallel to the clinical improvements reported in individuals with treatment-resistant depression following a single intravenous infusion (Abdallah et al., 2015). Of equal significance, ketamine failed to reverse the enhanced intolerance of uncertain threat induced by sgACC-25 over-activation as measured on the HI paradigm. This suggests symptom specificity in ketamine's action despite both symptoms being induced by the same over-activated brain region. Consequently, ketamine either acts differentially within area 25 to impact on appetitive but not aversive mechanisms or has differential effects on downstream output pathways of sgACC-25 related to reward, but not threat responsivity.

To determine the brain circuits that are altered by sgACC-25 over-activation, we combined intracerebral infusions, behavioural testing and $^{18}$F-fluorodeoxyglucose (FDG) PET imaging (Alexander et al., 2019, 2020). $^{18}$F-FDG is a glucose analogue, and uptake acts as a proxy measure of brain activity, with elevated uptake in regions that are more metabolically active. By manipulating the behavioural testing context – either appetitive or aversive – after an $^{18}$F-FDG injection but prior to imaging, we explored which functional circuits were engaged when sgACC-25 was over-activated in different contexts.

Over-activation of sgACC-25 in an appetitive context (exposure to Pavlovian conditioned stimuli predicting reward) elevated $^{18}$F-FDG uptake in the dorsomedial PFC (dmPFC), dorsal ACC (dACC) and insula, and reduced $^{18}$F-FDG uptake in regions of the brainstem, including the dorsal raphe nucleus, the medullary reticular formation and the nucleus of the solitary tract. By contrast, in an aversive context (exposure to Pavlovian conditioned stimuli predicting threat), sgACC-25 over-activation increased $^{18}$F-FDG uptake in the amygdala, hypothalamus and temporal association cortex, but decreased $^{18}$F-FDG uptake in the frontopolar cortex, dorsolateral PFC, central OFC and the caudate nucleus (Fig. 4).

The different circuits engaged may provide a mechanism by which ketamine's efficacious action

can ameliorate the blunting of reward anticipation, but not the heightened anxiety-like behaviour. In the appetitive context, we also imaged marmosets when ketamine was administered 1 day prior to sgACC-25 over-activation. Ketamine administration 1 day prior to over-activation/imaging normalized over-activation in the dmPFC, dACC and insula. By measuring $^{18}$F-FDG uptake within sgACC-25 itself in a region-of-interest analysis, it was evident that pre-treatment with ketamine 1 day prior to imaging also altered how the sgACC-25 responded to higher levels of extracellular glutamate induced by DHK infusions, possibly through neuroplastic changes within the region itself. The differential targets

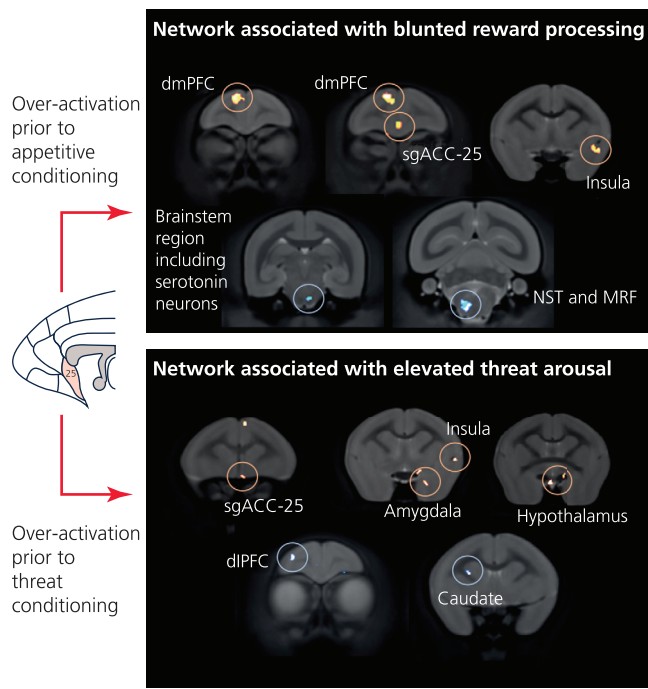

**Figure 4. Over-activation of marmoset sgACC-25 engages distinct networks in appetitive- and threat-related contexts, as measured by $^{18}$F-FDG PET imaging**
By over-activating sgACC-25 prior to $^{18}$F-FDG ligand injection and subsequent PET imaging, we could investigate the circuits engaged in contrasting appetitive and threatening contexts. If over-activated prior to exposure to an appetitive conditioned stimulus, there was increased $^{18}$F-FDG uptake in the dorsomedial (dm)PFC and insula, but reduced activity in a region of the brainstem including serotonergic neurons and in key autonomic centres including the nucleus of the solitary tract (NST) and the medullary reticular formation (MRF). However, if over-activated prior to exposure to a threat-associated conditioned stimulus, a distinct network was engaged including increased $^{18}$F-FDG uptake in the insula, amygdala and hypothalamus (together with the temporal association cortex, not shown), but reduced uptake in the dorsolateral (dl)PFC and caudate nucleus (together with the central OFC and frontopolar cortex, not shown).

for ketamine's action could be assessed in the future by studying the effects of local ketamine infusions.

## Appreciating the heterogeneity within the vmPFC

So far, the negative affective state induced by over-activation of marmoset sgACC-25, but not that of neighbouring pgACC-32, provides a causal link between increased activity in the caudal subcallosal cortex and negative affective responses in humans. But what of the neuroimaging in humans highlighting a region in more rostral vmPFC related to extinction recall of the threat response and more generally in inhibition of negative affective responses (Milad, Wright et al., 2007; Phelps et al., 2004)?

Area 32 inactivation in marmosets blunted the expression of conditioned threat and facilitated extinction, more comparable to a role in the inhibition of negative affective responses. Could this be the region in humans contributing to the imaging results? Certainly, a subcallosal part of area 32 lies more rostral than area 25 in humans. It should, however, be noted that area 32 manipulations in marmosets during threat extinction were restricted to the behavioural output, not impacting on the conditioned cardiovascular response, suggesting that the effect seen was not on the emotional response *per se* but instead on specific aspects of the behavioural orienting response to the CS, possibly reflecting attentional functions. Activity around the border of area 32/24 has, however, also been implicated in approach-avoidance decision making in macaque monkeys with neuronal activity representing motivationally positive and negative subjective value (Amemori & Graybiel, 2012). Less likely is contribution of neighbouring area 14 to this regulatory effect of suppressing negative affective responses since neither its activation nor its inactivation affects conditioned threat expression or extinction (Stawicka et al., 2020). Alternative regions responsible for the extinction effect in humans are medial area 10 (based on the parcellation of vmPFC by Ongür et al., 2003) or subcallosal area 24.

## Conclusion

Altogether, these results highlight several crucial translational considerations. First, cross-species comparisons of function should be interpreted with caution. We have shown that manipulations of the proposed anatomical homologues of IL and PL – sgACC-25 and pgACC-32, respectively – in the marmoset have opposite effects to those expected from the results of rodent studies with respect to the regulation of conditioned threat (but see Chen et al., 2021). However, in the context of conditioned reward, results do show considerable alignment,

with comparable blunting effects of DHK-induced over-activation on conditioned appetitive responses. These potential analogies and differences in cross-species comparisons across behavioural domains within a single region emphasize that future cross-species comparisons should focus on domain-specific neural circuits. For example, area 25 and IL in marmosets and rodents, respectively, may show greater analogy in their integration within appetitive circuits compared to threat-related circuits. Moreover, more emphasis should be placed on the investigation of pathway-specific projections from these regions and in primates on their interaction with higher-order dorsolateral and frontal polar regions. Second, symptom constructs in humans may reflect composite phenomena as a result of several neurobiological processes. This is particularly true when considering our results in the appetitive domain that are of relevance to anhedonia, which show that sgACC-25 over-activation blunts reward anticipation and motivation, but not reward consumption. Thus, it is important to compare across distinct psychological processes when determining the neural underpinnings of psychiatric symptoms. Finally, the complex and multi-faceted nature of emotion – consisting of subjective, behavioural and physiological components – means that the measurement of both behavioural and physiological parameters facilitates translation between animals and humans. This also provides insight into those prefrontal manipulations which impact the overall affective response as seen with area 25 interventions, influencing both cardiovascular and behavioural domains, compared to those involved in cognitive aspects of task performance as seen with area 32 interventions, which have little impact in the cardiovascular domain. Only closer integration of human, non-human primate and rodent studies will provide the necessary insight into the involvement of these brain regions at the intersection of cognition, physiology and behaviour.

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

## Additional information

### Competing interests

The authors declare no potential competing interests with respect to the research, authorship, and/or publication of this article.

### Author contributions

All authors contributed to the writing of this review and approved the final version of the manuscript. All authors have read and approved the final version of this manuscript and agree to be accountable for all aspects of the work in ensuring that questions related to the accuracy or integrity of any part of the work are appropriately investigated and resolved. All persons designated as authors qualify for authorship, and all those who qualify for authorship are listed.

### Funding

The marmoset work reviewed here was supported by an Investigator Award from the Wellcome Trust (108089/Z/15/Z) and a Medical Research Council programme grant to A.C.R. (M023990/1) and a Medical Research Council career development award (RG62920) to Dr Hannah Clarke.

### Acknowledgements

We thank in particular Drs Hannah Clarke, Chloe Wallis, Jorge Zeredo, Steve Sawiak, Tim Fryer and Young Hong, alongside Mrs Gemma Cockcroft and Mrs Lauren McIver, for their contribution to research presented in this review. Much of the marmoset work was conducted in the Behavioural and Clinical Neuroscience Institute.

### Keywords

anhedonia, anxiety, subcallosal cingulate

### Supporting information

Additional supporting information can be found online in the Supporting Information section at the end of the HTML view of the article. Supporting information files available:

**Peer Review History**

