## [Peer Review History · The Journal of Physiology]

Ventromedial Prefrontal Cortex and Emotion Regulation: Lost in Translation?

Angela C Roberts, Laith Alexander, and Christian M Wood
DOI: 10.1113/JP282627

Corresponding author(s): Angela Roberts (acr4@cam.ac.uk)

The following individual(s) involved in review of this submission have agreed to reveal their identity: Peter Rudebeck (Referee #1)

Review Timeline:

Submission Date:	30-Nov-2021
Editorial Decision:	04-Mar-2022
Revision Received:	10-May-2022
Accepted:	13-May-2022

Senior Editor: Ian Forsythe

Reviewing Editor: Michael Okun

Transaction Report:

Dear Dr Roberts,

Re: JP-SR-2021-282627 "Ventromedial Prefrontal Cortex and Emotion Regulation:Lost in Translation?" by Angela Roberts

Thank you for submitting your invited Review-Symposium to The Journal of Physiology. It has been assessed by a Reviewing Editor and by 1 expert referee and I am pleased to tell you that it is considered to be acceptable for publication following satisfactory revision.

The reports are copied at the end of this email. Please address all of the points and incorporate all requested revisions, or explain in your Response to Referees why a change has not been made.

NEW POLICY: In order to improve the transparency of its peer review process The Journal of Physiology publishes online as supporting information the peer review history of all articles accepted for publication. Readers will have access to decision letters, including all Editors' comments and referee reports, for each version of the manuscript and any author responses to peer review comments. Referees can decide whether or not they wish to be named on the peer review history document.

I hope you will find the comments helpful and have no difficulty in revising your manuscript within 4 weeks.

Your revised manuscript should be submitted online using the links in Author Tasks Link Not Available. This link is to the Corresponding Author's own account, if this will cause any problems when submitting the revised version please contact us.

The image files from the previous version are retained on the system. Please ensure you replace or remove any files that have been revised. Your revised submission should include:

- A Word file of the complete text (including figure legends any Tables);
- An Abstract Figure (with legend in the Article file)
- Each figure as a separate, high quality, file;
- A full Response to Referees;
- A copy of the manuscript with the changes highlighted.
- Author profile. A short biography (no more than 100 words for one author or 150 words in total for two authors) and a portrait photograph of the two leading authors on the paper. These should be uploaded, clearly labelled, with the manuscript submission. Any standard image format for the photograph is acceptable, but the resolution should be at least 300 dpi and preferably more.

- A 'Cover Art' file for consideration as the Issue's cover image;
- Appropriate Supporting Information (Video, audio or data set https://jp.msubmit.net/cgi-bin/main.plex?form_type=display_requirements#supp).

To create your 'Response to Referees' copy all the reports, including any comments from the Reviewing Editor into a Word, or similar, file and respond to each point in colour or CAPITALS and upload this when you submit your revision.

I look forward to receiving your revised submission.

If you have any queries please reply to this email and staff will be happy to assist.

Yours sincerely,

Ian D. Forsythe
Deputy Editor-in-Chief
The Journal of Physiology
<https://jp.msubmit.net>
<http://jp.physoc.org>
The Physiological Society
Hodgkin Huxley House
30 Farringdon Lane
London, EC1R 3AW
UK
<http://www.physoc.org>
<http://journals.physoc.org>

REQUIRED ITEMS:

-Your MS must include a complete "Additional information section" with the following 4 headings and content:

Competing Interests: A statement regarding competing interests. If there are no competing interests, a statement to this effect must be included. All authors should disclose any conflict of interest in accordance with journal policy.

Author contributions: Each author should take responsibility for a particular section of the study and have contributed to writing the paper. Acquisition of funding, administrative support or the collection of data alone does not justify authorship; these contributions to the study should be listed in the Acknowledgements. Additional information such as 'X and Y have contributed equally to this work' may be added as a footnote on the title page.

It must be stated that all authors approved the final version of the manuscript and that all persons designated as authors qualify for authorship, and all those who qualify for authorship are listed.

Funding: Authors must indicate all sources of funding, including grant numbers. If authors have not received funding, this must be stated.

It is the responsibility of authors funded by RCUK to adhere to their policy regarding funding sources and underlying research material. The policy requires funding information to be included within the acknowledgement section of a paper. Guidance on how to acknowledge funding information is provided by the Research Information Network. The policy also requires all research papers, if applicable, to include a statement on how any underlying research materials, such as data, samples or models, can be accessed. However, the policy does not require that the data must be made open. If there are considered to be good or compelling reasons to protect access to the data, for example commercial confidentiality or legitimate sensitivities around data derived from potentially identifiable human participants, these should be included in the statement.

Acknowledgements: Acknowledgements should be the minimum consistent with courtesy. The wording of acknowledgements of scientific assistance or advice must have been seen and approved by the persons concerned. This section should not include details of funding.

-Please upload separate high quality figure files via the submission form.

-Author profile(s) must be uploaded via the submission form. Authors should submit a short biography (no more than 100 words for one author or 150 words in total for two authors) and a portrait photograph of the two leading authors on the paper. These should be uploaded, clearly labelled, with the manuscript submission. Any standard image format for the photograph is acceptable, but the resolution should be at least 300 dpi and preferably more. A group photograph of all authors is also acceptable, providing the biography for the whole group does not exceed 150 words.

EDITOR COMMENTS

Reviewing Editor:

Thank you for submitting the review manuscript to Journal of Physiology. I would like to apologise for the excessive time that the review process has taken. We solicited two reviews of your submission, however we lost contact with one of the reviewers, who accepted the review invitation but did not respond to any of our subsequent emails. This is also the reason for the delay in our response.

The opinion of the other reviewer was highly favourable. This reviewer provided several detailed suggestions for improving the manuscript.

Senior Editor:

Thank you for an interesting review. In responding to the referee comments please consider how to make your article appeal to the widest audience.

REFeree COMMENTS

Referee #1:

This review by Alexander, Wood and Roberts covers a number of recent high-profile reports from this laboratory on the functions of the medial frontal cortex in marmosets with a specific focus on regulation of autonomic responses. The top-level message is that non-human primates and marmosets in particular are an important translational model for neuroscience is important and this manuscript is well motivated. The review aims to thoroughly summarise the current knowledge on cross-species similarities and discrepancies in ventromedial frontal cortex anatomy, peripheral physiology, and behavior-related activity in rodents, a new world-monkey species (marmosets), macaque monkeys and humans. It specifically focuses on the comparative role of the primate subcallosal area 25 (infralimbic area in rodents) and cingulate area 32 (prelimbic area in rodents). Then it goes onto more translational issues related to the impact of ketamine on affect and how pre-treatments with ketamine can block alterations in emotion caused by overactivation of area 25 in marmosets.

On the whole, the manuscript presents a comprehensive enough review of the current literature on this topic and nicely explains the current discrepancies in results across primates and rodents. On the other hand, to fully accomplish this goal and strengthen the potential impact of this manuscript, there are a number of places where the authors need to do some work. These issues mostly relate to connecting to the prior work on areas 25 and 32 in macaques as well as comparative neuroanatomy. One point that it feels like the authors should note somewhere in their manuscript is that while much of their excellent neuropsychological work has been conducted in marmosets, they draw heavily on anatomy and physiology work conducted in macaques. Indeed, the work cited on the connections of non-human primate medial frontal cortex almost exclusively references studies in macaques. As statement saying as such should be included. In addition, specific comments are below.

1. The brief, but essential discussion of comparative anatomy at the start of the manuscript would be enhanced by including a figure showing the different areas in humans, marmosets and rodents that are being referred to. This would help a reader unfamiliar with the location of these areas as well as highlight the similarity in location across species. At present the areas are shown in the graphical abstract but not in a main figure. In this context, it also might be worth noting that the parcellation of human medial frontal cortex differs between researchers. For instance, the maps of Petrides and Pandya differs from those of Ongur, Price, and colleagues. These differences between maps are especially important for determining homologies from humans to non-human primates especially regarding the imaging results from humans that the authors discuss.

2. The authors mention area 33 in the first section of the manuscript stating without reference that this area is involved interoceptive processes. They then drop the area from further discussion without reason. It feels like they should either do a more in-depth discussion of the comparative anatomy and function of area 33 or consider dropping referring to this area from the manuscript altogether.

3. On page 5, the authors briefly describe inter-species differences and similarities of area 25 connections with other parts of the brain (i.e. auditory cortex, nucleus of the solitary tract and other frontal areas). Considering the time spent on discussing functional differences in autonomic-related behaviors between rodents and primates, more detail on the connections between area 25/32 with amygdala, insular cortex and OFC as well as brain stem areas across non-human primates and rodents is needed here. In particular the current level of detail is quite superficial and only recent reports in macaques from Joyce and Barbas as well as Heilbronner and Haber are cited when there are many, many other papers. For instance, Freedman et al., JCN (2000) characterized the connections of macaque area 25 but there is no reference to this paper, as are resources that detail the connections of medial frontal areas in marmosets, (see Majka et al., 2020). Furthermore, a brief discussion of differences/similarities in intra-cingulate connectivity (especially across area 25 and 32) across these species may be relevant here.

4. The review focuses extensively on work tying together manipulations of neural activity with assessments of behavior and cardiovascular/autonomic activity in the different species. There are a number of comments here.

First, I was wondering whether the authors know if the cardiovascular system (and not just the neural activity) in marmosets is comparable to the one in rodents, old-world monkeys and humans. A paragraph on homologies in the cardiovascular system (and its link to cognitive processes) feels appropriate here?

Second, the authors work has used pharmacological agents to manipulate activity in medial frontal cortex to impact cardiovascular responses in marmosets. An extensive literature in humans, macaques and rodents has detailed how electrical stimulation of different medial frontal areas impacts heart rate, breathing, blood pressure as well as other measures of autonomic function. That work should be included here as it helps set the stage for the author's work. Here I'm thinking of older work from Kaada in macaques, as well as Smith, Ward, Lewin and others in humans.

5. The umbrella term of vmPFC is very appropriate in the context of primates. However, due to the differences in cytoarchitecture across primates and rodents in some frontal regions, I wonder whether it may be more appropriate to leave the "prefrontal" component out and simply refer to frontal cortex throughout the manuscript. I would suggest to change vmPFC into ventromedial frontal cortex (vmFC), especially considering the goal of the review is compare this primate part of the frontal cortex with the rodents one.

6. The review would benefit from including a table to summarize the results of over-activation/disruption of area 25/IL in rodents, marmosets and macaques by placing the three species one next to the other.

7. In a number of places throughout the manuscript references are missing when a statement of fact is made. Here are two examples but there are more:

"In addition, both subgenual area 24 and area 33 were associated with interoception: subgenual 24 with taste and 33 with pain."

"This is contributed to, in part, by signal dropout in this region in neuroimaging studies, making it difficult to be confident about the precise location of the activation."

END OF COMMENTS

Confidential Review

30-Nov-2021

We are delighted to hear Referee 1 felt our review was comprehensive. We address Referee 1's comments below.

1. The brief, but essential discussion of comparative anatomy at the start of the manuscript would be enhanced by including a figure showing the different areas in humans, marmosets and rodents that are being referred to. This would help a reader unfamiliar with the location of these areas as well as highlight the similarity in location across species. At present the areas are shown in the graphical abstract but not in a main figure. In this context, it also might be worth noting that the parcellation of human medial frontal cortex differs between researchers. For instance, the maps of Petrides and Pandya differs from those of Ongur, Price, and colleagues. These differences between maps are especially important for determining homologies from humans to non-human primates especially regarding the imaging results from humans that the authors discuss.

We agree with this comment and have in other reviews highlighted the differences between maps but didn't here. However, we agree that we should, so we have now highlighted these differences in the revised version and include maps across species in the main manuscript.

"Of note are the regional differences in the maps of Ongür *et al.* and Petrides and Pandya, particularly with regards the extent of areas 14 and 10 on the caudal aspects of the medial surface, and the extension (or not) of area 25 onto the orbital surface. This further complicates the determination of homologies between species."

2. The authors mention area 33 in the first section of the manuscript stating without reference that this area is involved interoceptive processes. They then drop the area from further discussion without reason. It feels like they should either do a more in-depth discussion of the comparative anatomy and function of area 33 or consider dropping referring to this area from the manuscript altogether.

We agree with this comment. We have decided to remove reference to area 33 as it is infrequently described in anatomical maps of the subgenual region.

3. On page 5, the authors briefly describe inter-species differences and similarities of area 25 connections with other parts of the brain (i.e. auditory cortex, nucleus of the solitary tract and other frontal areas). Considering the time spent on discussing functional differences in autonomic-related behaviors between rodents and primates, more detail on the connections between area 25/32 with amygdala, insular cortex and OFC as well as brain stem areas across non-human primates and rodents is needed here. In particular the current level of detail is quite superficial and only recent reports in macaques from Joyce and Barbas as well as Heilbronner and Haber are cited when there are many, many other papers. For instance, Freedman *et al.*, JCN (2000) characterized the connections of macaque area 25 but there is no reference to this paper, as are resources that detail the connections of medial frontal areas in marmosets, (see Majka *et al.*, 2020). Furthermore, a brief discussion of differences/similarities in intra-cingulate connectivity (especially across area 25 and 32) across these species may be relevant here.

A thorough comparison of area 25 /IL was performed in a previous review (Alexander *et al.*, 2019) that included many anatomical papers from both rats and macaques which we did not want to duplicate in this review. Hence why we provided a brief summary and referred people to that review. However, we did not provide a detailed comparison of area 32 and have not included Majka *et al* 2020 although the latter only includes an injection into area 32 not area 25, unfortunately. However, we have now included a more detailed discussion about the similarities and differences between primate area 25 and rodent IL and primate area 32 and rodent PL.

"Consistent with their respective homologies, there are marked similarities in the connectivity patterns of rodent IL and non-human primate area 25, and rodent PL and non-human primate area 32. In the case of IL and area 25, connectivity with regions linked to emotion, visceromotor control and memory is broadly similar. Subcortically, this includes the hypothalamus, amygdala, BNST, lateral septum, nucleus accumbens, midline thalamic nuclei, periaqueductal grey (PAG) and parabrachial nucleus; and cortically, this includes the medial orbitofrontal cortex (OFC), insula, anterior cingulate, perirhinal cortex, entorhinal cortex, hippocampal formation and PL/area 32 (Vertes, 2004; Joyce & Barbas, 2018). Regarding PL and area 32, these regions project to the amygdala, hypothalamus, medial caudate and nucleus accumbens subcortically; and the agranular insular, other cingulate subregions, and OFC cortically (Chiba *et al*, 2001; Yeterian *et al*, 2012).

However, cross-species differences in connectivity do exist, especially when considering cortico-cortical connectivity. Absent in rodent IL, but present in macaque area 25, are connections with auditory association and polymodal sensory cortex within the superior temporal gyrus (Vertes, 2004; Hoover & Vertes, 2007). In addition, rodent IL lacks the prominent connections that primate area 25 has with higher-order associations areas within PFC, including area 9, 10 and 46 and posterior cingulate area 23, regions with no known anatomical homology in

rodents (compare (Yeterian et al., 2012) and (Joyce & Barbas, 2018) with (Vertes, 2004) and (Hoover & Vertes, 2007)).

In addition, even subcortically where there is considerable comparability, there are notable differences in the extent and the specificity of connectivity within regions. First, macaque area 25 neurons project primarily to central and basal amygdala nuclei. In contrast, a major projection of IL is onto the GABAergic inhibitory intercalated cell masses (proposed to mediate the inhibitory effects of IL stimulation on conditioned threat responses- see below), which is more comparable to connections of posterior OFC in macaques (Joyce & Barbas, 2018). Second, connectivity of primate area 25 is more extensive and widespread across the hypothalamus compared to that of rodent IL (Rempel-Clower & Barbas, 1998; Ongür et al., 1998; Vertes, 2004), with many more hypothalamic neurons sending reciprocal projections back to macaque area 25. Third, in the brainstem, primate area 25 projects predominantly to dIPAG whilst rodent IL projects more evenly to both vIPAG and dIPAG; vIPAG associated with passive coping responses whilst dIPAG with active coping responses (An et al., 1998; Floyd et al., 2000). Moreover, absent in macaque area 25, but prominent in rodent IL, are direct projections to autonomic effectors including the nucleus of the solitary tract, ventrolateral medulla and intermediolateral cell column (Joyce & Barbas, 2018). Area 25 is likely to have effects instead indirectly via the parabrachial nucleus and forebrain sites including the hypothalamus and BNST.

Like IL and area 25, differences between PL and area 32 are most striking when considering their cortical connectivity patterns. First, pgACC-32 has far greater connectivity with lateral prefrontal association areas (8, 9, 46 and 47), premotor area 6 as well as posterior cingulate regions, 23 and 31, and retrosplenial cortices, 29 and 30, than neighboring sgACC-25, but aside from premotor and retrosplenial cortex, these regions have no known structural homologs in rodents (compare (Chiba et al., 2001; Yeterian et al., 2012) with (Vertes, 2004; Hoover & Vertes, 2007)). Like sgACC-25, pgACC-32 also has extensive connections with the auditory association and multimodal sensory regions of the STG, projections that are absent in rodent PL; these temporal regions projecting instead to more dorsal cingulate regions in rodents (Hoover and Vertes, 2007).

Thus, whilst there is considerable overlap in connectivity between rat IL and monkey area 25, and rat PL and monkey area 32, the cortical connectivity patterns of these two brain regions in macaque set them apart from their proposed rodent counterparts. Consequently, whilst they may be involved in similar subcortical functional networks across species, the precise role they play may well have altered.”

4. The review focuses extensively on work tying together manipulations of neural activity with assessments of behavior and cardiovascular/autonomic activity in the different species. There are a number of comments here.

First, I was wondering whether the authors know if the cardiovascular system (and not just the neural activity) in marmosets is comparable to the one in rodents, old-world monkeys and humans. A paragraph on homologies in the cardiovascular system (and its link to cognitive processes) feels appropriate here? Second, the authors work has used pharmacological agents to manipulate activity in medial frontal cortex to impact cardiovascular responses in marmosets. An extensive literature in humans, macaques and rodents has detailed how electrical stimulation of different medial frontal areas impacts heart rate, breathing, blood pressure as well as other measures of autonomic function. That work should be included here as it helps set the stage for the author's work. Here I'm thinking of older work from Kaada in macaques, as well as Smith, Ward, Lewin and others in humans.

We have added more detail to the homologies related to the cortical control of cardiovascular function, together with reference to work in macaques, in the following paragraph:

“The profound consequences of sgACC-25 inactivation on resting autonomic tone have apparent, albeit tenuous, similarity to manipulations of its rodent counterpart. Rodent IL (and ventral PL) has been termed ‘visceral motor cortex’ with projections well placed to alter autonomic tone, including to the hypothalamus, amygdala, insula and peri-aqueductal gray (Loewy & Spyer, 1990; Vertes, 2004). Both inactivation and activation of IL has cardiovascular effects, although the directionality of these effects is variable (Terreberry & Neafsey, 1983; Loewy & Spyer, 1990; Tavares *et al.*, 2009; Müller-Ribeiro *et al.*, 2012; Hassan *et al.*, 2013). Macaque vmPFC extensively projects to regions involved in the central regulation of autonomic function (Barbas *et al.*, 2003); pioneering work by Kaada and colleagues showed that direct electrical stimulation of macaque pgACC and sgACC induces cardiovascular and respiratory changes, which were most pronounced following stimulation of caudal regions corresponding to sgACC-25 (Kaada *et al.*, 1949). However, electrical stimulation can impact on axons passing through the targeted region and so the localizability of these findings to the target region remains unclear. Selective stimulation /inactivation of cell bodies in marmosets confirm the specific involvement of sgACC-25 in autonomic regulation and map closely onto results from human neuroimaging

studies exploring the neural correlates of heart rate variability – a measure reflecting dynamic adjustments in heart rate mediated by the parasympathetic and sympathetic nervous systems. These studies suggest that human sgACC-25 BOLD activity correlates closely with changes in parasympathetic tone during affective state switching (Lane *et al.*, 2013), supporting the profound effects of marmoset sgACC-25 inactivation on cardiac vagal tone.”

The work of Ward, Smith, Lewin – whilst informative – is not included in this review. The papers either do not clearly delineate the region of the anterior cingulate stimulated, or, stimulate human dorsal/mid ACC rather than pgACC-32 or sgACC-25. The work by Kaada stimulates macaque sgACC-25 and is therefore included.

5. The umbrella term of vmPFC is very appropriate in the context of primates. However, due to the differences in cytoarchitecture across primates and rodents in some frontal regions, I wonder whether it may be more appropriate to leave the "prefrontal" component out and simply refer to frontal cortex throughout the manuscript. I would suggest to change vmPFC into ventromedial frontal cortex (vmFC), especially considering the goal of the review is compare this primate part of the frontal cortex with the rodents one.

We absolutely take on board the issue that the reviewer is raising: cingulate regions are not strictly speaking part of prefrontal cortex. However, as the starting point of this review is vmPFC in humans, a term that has traditionally and is still currently used to refer to the entire region, including the caudal cingulate regions, we are loathed to remove this term altogether and replace it with vmFC. However, to acknowledge this issue which we agree is an important one, we now speak to it in the first section in which we begin to talk about cross-species comparison, under the heading 'lost in translation'. Specifically:

“It has been hypothesized that anatomically homologous regions within vmPFC are present across rodents, monkeys and humans which should afford the opportunity to establish causal evidence for functional heterogeneity suggested by the neuroimaging literature in humans. It is important to recognize though that whilst the term vmPFC is commonly used to refer to the entire ventromedial frontal lobes, only more rostral regions are specifically prefrontal. More caudal aspects are part of the cingulate cortex, yet the proposed anatomical cross-species homology relates primarily to these cingulate regions of 24, 25 and 32.”

6. The review would benefit from including a table to summarize the results of over-activation/disruption of area 25/IL in rodents, marmosets and macaques by placing the three species one next to the other.

We considered creating a table but after a few attempts we decided that the table was not useful. There are so few studies where the same behavioural domain and/or same manipulation has been used across species to warrant it.

7. In a number of places throughout the manuscript references are missing when a statement of fact is made.

We have read through the manuscript and included references where these are missing.

Dear Dr Roberts,

Re: JP-SR-2022-282627R1 "Ventromedial Prefrontal Cortex and Emotion Regulation:Lost in Translation?" by Angela C Roberts
Laith Alexander
Christian M Wood

I am pleased to tell you that your Symposium Review article has been accepted for publication in The Journal of Physiology, subject to any modifications to the text that may be required by the Journal Office to conform to House rules.

NEW POLICY: In order to improve the transparency of its peer review process The Journal of Physiology publishes online as supporting information the peer review history of all articles accepted for publication. Readers will have access to decision letters, including all Editors' comments and referee reports, for each version of the manuscript and any author responses to peer review comments. Referees can decide whether or not they wish to be named on the peer review history document.

The last Word version of the paper submitted will be used by the Production Editors to prepare your proof. When this is ready you will receive an email containing a link to Wiley's Online Proofing System. The proof should be checked and corrected as quickly as possible.

All queries at proof stage should be sent to tjp@wiley.com

The accepted version of the manuscript is the version that will be published online until the copy edited and typeset version is available. Authors should note that it is too late at this point to offer corrections prior to proofing. Major corrections at proof stage, such as changes to figures, will be referred to the Reviewing Editor for approval before they can be incorporated. Only minor changes, such as to style and consistency, should be made a proof stage. Changes that need to be made after proof stage will usually require a formal correction notice.

Are you on Twitter? Once your paper is online, why not share your achievement with your followers. Please tag The Journal (@jphysiol) in any tweets and we will share your accepted paper with our 22,000+ followers!

Yours sincerely,

Ian D. Forsythe
Deputy Editor-in-Chief
The Journal of Physiology
<https://jp.msubmit.net>
<http://jp.physoc.org>
The Physiological Society
Hodgkin Huxley House
30 Farringdon Lane
London, EC1R 3AW
UK
<http://www.physoc.org>
<http://journals.physoc.org>

Comments:

Reviewing Editor:

Thank you for addressing the points raised by the reviewer in the revised version of your submission.

Senior Editor:

Congratulations on an interesting and informative review!

REFEREE COMMENTS:

Referee #1:

The authors have done an excellent job responding to my comments and this is now a particularly informative and useful review of the literature and the ground-breaking work of this group.

While I could quibble about their use of the term "prefrontal" in the manuscript, the authors have appropriately highlighted the controversy around this point in the revised version. This is more than sufficient.

* IMPORTANT NOTICE ABOUT OPEN ACCESS *

To assist authors whose funding agencies mandate public access to published research findings sooner than 12 months after publication The Journal of Physiology allows authors to pay an open access (OA) fee to have their papers made freely available immediately on publication.

You will receive an email from Wiley with details on how to register or log-in to Wiley Authors Services where you will be able to place an OnlineOpen order.

You can check if you funder or institution has a Wiley Open Access Account here <https://authorservices.wiley.com/author-resources/Journal-Authors/licensing-and-open-access/open-access/author-compliance-tool.html>

Your article will be made Open Access upon publication, or as soon as payment is received.

If you wish to put your paper on an OA website such as PMC or UKPMC or your institutional repository within 12 months of publication you must pay the open access fee, which covers the cost of publication.

OnlineOpen articles are deposited in PubMed Central (PMC) and PMC mirror sites. Authors of OnlineOpen articles are permitted to post the final, published PDF of their article on a website, institutional repository, or other free public server, immediately on publication.

Note to NIH-funded authors: The Journal of Physiology is published on PMC 12 months after publication, NIH-funded authors DO NOT NEED to pay to publish and DO NOT NEED to post their accepted papers on PMC.

1st Confidential Review

10-May-2022